# Click-to-Release for Controlled Immune Cell Activation: Tumor-Targeted Unmasking of an IL12 Prodrug

**DOI:** 10.3390/ph18091380

**Published:** 2025-09-16

**Authors:** Martijn H. den Brok, Kim E. de Roode, Luc H. M. Zijlmans, Laurens H. J. Kleijn, Marleen H. M. E. van Stevendaal, Ron M. Versteegen, Lieke W. M. Wouters, Raffaella Rossin, Marc S. Robillard

**Affiliations:** 1Tagworks Pharmaceuticals, Toernooiveld 1, 6525 ED Nijmegen, The Netherlands; 2Department of Medical Imaging, Nuclear Medicine, Radboud UMC, Geert Grooteplein Zuid 10, 6525 GA Nijmegen, The Netherlands; 3SyMO-Chem, Den Dolech 2, 5612 AZ Eindhoven, The Netherlands

**Keywords:** IL12, cytokine, immunotherapy, cancer, masking, polyethyleneglycol, click, bioorthogonal, click-to-release, *trans*-cyclooctene, tetrazine, unmasking

## Abstract

**Objectives:** Immunotherapy utilizing immune-stimulating cytokines such as IL12 holds great promise for the treatment of cancer. However, clinical use of IL12 is hampered due to severe toxicity following systemic administration. We here present a novel treatment strategy in which IL12 is chemically silenced by conjugation to PEG masks that sterically hinder the receptor binding. Subsequently, the masks can be released on demand using a bioorthogonal click reaction, cleaving the linker connecting the masks, thereby restoring the native cytokine. This “click-to-release” approach is based on the highly selective Inverse electron-demand Diels–Alder (IEDDA) pyridazine elimination reaction between a tetrazine (Tz) and a *trans*-cyclooctene (TCO), optimized for fast reaction kinetics and in vivo compatibility. Selective activation in the tumor microenvironment is achieved by pretargeting one component of this reaction to the tumor, triggering local activation of the masked IL12 once it is given in a secondary i.v. injection. **Methods:** IL12 masking and unmasking were evaluated in vitro with PAGE and HEK-Blue reporter cells and ex vivo with ELISA. Biodistribution in mice was evaluated with I-125 radiolabeling and biotin-click histochemistry. **Results:** Several designs were evaluated and optimized in vitro, resulting in an IL12-TCO-PEG construct that exhibited superior masking and subsequent reactivation upon reaction with a tetrazine bound to a TAG-72-targeted diabody. In tumor-bearing mice, we demonstrated that this diabody-tetrazine could efficiently pre-localize tetrazine in the tumor. Administration of IL12-TCO-PEG 24 h later afforded efficient and selective unmasking in tumors, but not in the blood. **Conclusions:** These results demonstrate proof of principle of the click-cleavable IL12 prodrug approach and showcase the versatility of the click-to-release reaction.

## 1. Introduction

The development of next-generation immunotherapies involving recombinant cytokines or chemokines holds great promise for the treatment of cancer [1,2,3]. These therapies were shown to recruit effector immune cells to the often-immunosuppressive local tumor microenvironment (TME), thereby converting immunologically ‘cold’ tumors into ‘hot’ ones, resulting in tumor responses in the clinic. Interleukin-12 (IL12) is a key regulator of innate and adaptive cell-mediated immunity, responsible for multiple anti-tumor effects [4,5]. It stimulates interferon-γ (IFN-γ) production and supports the cytolytic potential of various immune effector cells [6,7]. Furthermore, IL12 promotes the differentiation of CD4+ T cells into T helper 1 (Th1) cells and enhances antibody-dependent cellular toxicity [2,4,8]. In preclinical therapy studies IL12 has been shown to generate robust antitumor activity [6,8], which was confirmed to a limited extent in patients treated with systemically injected recombinant cytokine [8]. Unfortunately, the clinical use of IL12 has been largely impeded by its short half-life, the severe toxicity associated with its systemic administration, and limited efficacy at the maximum tolerated dose [8,9,10].

Local delivery of IL12 in or close to the TME would allow higher dosages by significantly limiting toxicity. To achieve this, several strategies have been applied, aiming to either locally transcribe IL12 or to target it to the tumor [9]. However, intratumoral injection of IL12-encoding RNA or DNA in nanoparticles suffers from unpredictable expression levels, genomic integration of DNA, and the often impractical administration route [11,12,13,14]. To achieve systemic tumor targeting of IL12, various cytokine-antibody fusions were investigated (pre)clinically [15,16,17,18,19,20], but since these immunocytokines retain their activity, the systemic toxicity is not abrogated. Masked cytokines have been developed to be activated by slow hydrolysis or by more selective protease-based cleavage in the TME. Masking could be achieved by conjugating polyethylene glycol (PEG) chains to the cytokine [21,22] or, in the case of IL12, by using part of the IL12 receptor as a blocking domain [23,24,25]. However, relying on overexpressed proteases in the TME to release the masks may be hampered by TME variability and unintended off-target activity.

To overcome the aforementioned challenges, we set out to develop activatable proteins that are responsive to bioorthogonal chemistry instead of biological mechanisms. We previously developed bioorthogonal cleavage reactions based on the inverse electron-demand Diels-Alder (IEDDA) click reaction between a tetrazine (Tz) and *trans*-cyclooctene (TCO). In these IEDDA pyridazine elimination reactions, either a TCO-linked payload is cleaved by a Tz trigger or vice versa, a Tz-linked payload is cleaved by a TCO trigger, in both cases converting a carbamate-linked payload to a free amine-containing payload [26,27]. These so-called “click-to-release” reactions have led to many in vivo applications, such as Tz-triggered on-target cleavage and activation of TCO-containing antibody-drug conjugates (ADCs), prodrugs, and proteins, as well as the off-target deactivation of radioimmunoconjugates [28,29,30].

We here report on the development of click-activatable IL12 cytokine constructs, masked by conjugation of multiple lysine residues to PEG chains, connected via TCO or Tz- click-to-release linkers (Figure 1). Upon reaction with the trigger component, the PEG masks are designed to be tracelessly released, returning a native human IL12 protein with full bioactivity. To achieve tumor-selective activation of the click-cleavable masked cytokine, the trigger was pretargeted to tumors by a TAG-72-directed diabody, which binds LS174T colorectal tumor xenografts in mice with a high tumor-to-blood ratio. The masked IL12 was administered in a second step 24 h later, when the trigger had cleared sufficiently from circulation. This resulted in remarkably efficient and complete unmasking of IL12 in the tumor, while avoiding IL12 activation in the blood. This proof of principle of chemically triggered on-tumor IL12 activation represents a promising and differentiated approach compared to previous IL12 delivery methods in that it may afford systemic administration of an inactive IL12 construct combined with selective and predictable chemical activation in the tumor instead of relying on variable TME biology.

## 2. Results

The human IL12 protein harbors two subunits. When examining the sequence of the p40 subunit (UniProt P29460, residues 23–328), we noted that it contained a large number of lysine residues (26) that appeared to be surface-exposed in the area surrounding the binding domain with the IL12 receptor (IL12Rβ1) [31]. We therefore hypothesized that a lysine-based masking strategy could work particularly well to silence IL12 activity. To create an IL12 cytokine that would have strongly diminished systemic bioactivity in vivo, we developed masks based on relatively short PEG chains comprising a Tz or TCO linker and an amine-reactive pentafluorophenyl (PFP) carbonate (Figure 2A and Figure 3A). PEG conjugation to lysine residues in IL12 affords a carbamate bond, which can be cleaved with trigger reforming the native lysine through a traceless IEDDA pyridazine elimination reaction, restoring full bioactivity of native IL12 in a controlled manner. Since TCOs have been extensively used as mAb-tag in pretargeting [32], as a first design, we joined the PEG masks to IL12 via Tz linkers and employed TCO as the pretargeted trigger (Figure 2A). For this, we utilized the previously described click-cleavable tetrazine-carbamate linkage [27]. Based on this earlier work, we expected that direct lysine conjugation of PEG-Tz through a methylene-linked carbamate would result in an undesired destabilizing effect from the primary carbamate NH on the tetrazine. Therefore, we incorporated an N-methyl-substituted para-aminobenzyloxycarbonyl (MePABC) linker between the Tz and the cytokine, which, upon reaction with a TCO trigger, would self-immolate and liberate the IL12 lysine.

Conjugation of PFP-carbonate-activated PABC-Tz-PEG mask **1** (Figure 2A,B) to IL12 resulted in an 80 kDa product (**4**), with an approximate mask-to-protein functionalization grade (FG) of 10 (Figure 2C, lane 3). To evaluate the masking, a bioactivity assay was performed using HEK-Blue reporter cells, which carry the human genes for the IL12 receptor, its signaling pathway, and a STAT4 inducible SEAP reporter gene. Figure 2D shows that bioactivity was 10-fold reduced after conjugation, indicating partial silencing of IL12. To ensure IL12 is unmasked in the tumor only, and not in circulation, the pretargeting antibody-TCO conjugate should be cleared from circulation by the time the masked IL12 is administered. We therefore site-specifically conjugated four copies of TCO trigger **2** (Figure 2A) to a rapidly clearing diabody (t_1/2_ < 5 h) derived from the murine IgG CC49, targeting the non-internalizing pan-carcinoma glyco-epitope TAG-72 [33] (**5**; Figure 2B,C, lane 6). TAG-72 is a pan-carcinoma glycoepitope that was first identified by Colcher et al. [34,35]. It is highly expressed in a broad range of cancers, predominantly (adeno)carcinomas, including endometrial, ovarian, prostatic, colorectal, and gastric cancers. The TAG-72 targeting IgG CC49 and the diabody derivative have been successfully used for selective tumor (pre)targeting in the clinic [33,36]. Furthermore, the fast clearance and non/slow-internalizing nature of this diabody has been used in 2-step targeting and activation of a click-cleavable ADC [28], promising sufficient extracellular exposure of the trigger at the moment of masked IL12 injection in the present study. Unfortunately, the reaction of diabody-TCO **5** did not seem to activate IL12 construct **4**, while non-conjugated TCO **3** afforded only partial activation in the bioactivity assay (Figure 2D). This was reflected in the PAGE gels of the reaction mixtures of radiolabeled masked IL12 **4** and diabody-TCO **5** or TCO **3** (Figure 2C, lanes 8 and 4, respectively). This incomplete activation is likely the result of the fact that the tetrazine-carbamate cleavage gives high but not quantitative release yields (ca. 90%) [27] which results in a substantial cumulative fraction of non-activated IL12 when 10 lysines are masked. In addition, a clicked but not released diabody would further obstruct and mask the IL12 due to its large size.

Recently, Kuba, et al., developed a TCO that affords quantitative release with a wide range of tetrazines due to a hydroxyl substituent that facilitates the formation of the releasing dihydropyridazine intermediate [37]. This led us to evaluate a reverse approach in which a TCO is used as the linker in the masking moiety and Tz as the trigger for release (Figure 3A). The accessible lysine residues in IL12 were conjugated to two different PFP-activated TCO masks comprising PEG_2_ (**6**) or PEG_1k_ (**7**) moieties, affording IL12 conjugates **10** and **11** (Figure 3B). Using increasing reaction equivalents of PFP-TCO-PEG_2_ and PFP-TCO-PEG_1K_ masks resulted in an incrementally increased functionalization grade FG ranging from 7-30 and 3-13 for PEG_2_ and PEG_1K,_ respectively (Figure 3C,D). Using PEG_1K_ masks resulted in a ca. 2.5-fold lower conjugation efficiency and number of masks per molecule, probably related to the increased bulkiness of the PEG_1K_ chains. Size exclusion chromatography demonstrated good linker and protein stability of the constructs over several weeks at 4 °C (data not shown). IL12 bioactivity was greatly diminished for both the PEG_2_ and PEG_1K_-masked constructs, showing complete blocking at higher FG (Figure 3E). Based on the reduced blocking at lower FG of the PEG_1K_ construct, we concluded that the minimum FG for complete masking lies between 9 and 13. Importantly, upon adding merely five molar equivalents of highly reactive bis-2-pyridyl-tetrazine (**8**) relative to TCO, the masks were efficiently and fully released, and the apparent mass of native IL12 was regained for all constructs (Figure 3C), leading to full recovery of bioactivity to the level of native IL12 (Figure 3E). As both masks were effective, we continued the conjugate optimization with the less bulky PEG_2_ mask. We selected an FG of ca. 11 (20 carbonate-protein reaction equivalents) for the subsequent evaluations as we believed this would afford sufficient masking while avoiding excessive IL12 functionalization that could hamper on-tumor activation efficiency (Figure 3F).

Although bis-2-pyridyl-tetrazines are often used as fast-clearing pretargeting agents because of their high reactivity, their resulting limited stability [38] makes them less suitable as pretargeted mAb-tags, which require a far longer in vivo residence time. Recently, Svatunek et al. reported the development of cyclic vinyl ether-substituted tetrazines in which N-O repulsion increases reactivity without reducing stability [39]. The bis-vinyl ether Tz was found to be sufficiently reactive towards TCO while having superior stability to bis-2-pyridyl-Tz, which would make it suitable for extended in vivo exposure as a pretargeting tag. We therefore prepared a bis-vinyl ether Tz with a maleimide handle for cysteine conjugation (**9**, Figure 3A), and the corresponding diabody-tetrazine conjugate with a functionalization ratio of 4 (**12**). Next, we evaluated the activation of TCO-PEG_2_-masked IL12 (**10**; FG 11) using this construct in vitro and were gratified to find that, similar to Tz **8,** the diabody-Tz could efficiently react with all the TCO-linked PEG_2_ masks in the IL12 construct, returning a fully active native IL12 (Figure 3F).

We next set out to evaluate the concept in mice. To be able to analyze masked and unmasked IL12 in murine plasma samples, we investigated whether it could be detected by a standard IL12 sandwich-ELISA. While unmasked IL12 constructs were fully detectable, both the PEG_2_- and PEG_1K_-masked constructs were not, suggesting that the binding spot of (at least one of) the ELISA’s antibodies was blocked by masking (Figure 3G). We therefore concluded that this ELISA was a valid method for ex vivo detection of IL12 unmasking. Blood kinetic analysis of ^125^I-labeled IL12-TCO-PEG_2_ (**10**) showed that it had a similarly fast clearance as native IL12, with <0.5%ID/g left in blood at 24 h (Figure 4A). Similarly, blood kinetic analysis of ^125^I-labeled diabody-Tz (**12**) showed that after 24 h, its presence in circulation had diminished to <0.5%ID/g (Figure 4B). The diabody-Tz distributed efficiently to TAG-72 positive LS174T tumors, as demonstrated in a mouse biodistribution experiment (Figure 4C). Using TCO-biotin click histochemistry on the tumors ex vivo, we could demonstrate that reactive Tz was localized to the tumor (Figure 4D). Furthermore, autoradiographic imaging of iodine-125 in whole tumor mounts overlapped with histochemistry staining, demonstrating colocalization of the diabody and the Tz inside the tumor (Figure 4E).

To obtain in vivo proof of locally unmasked IL12 in the tumor, diabody-Tz conjugate **12** was injected i.v. in LS174T tumor-bearing mice, followed by injection of the IL12-TCO-PEG_2_ **10** 24 h later when **12** had cleared from circulation. As the IL12 construct had a short circulation combined with an expected fast and continuous local IL12 unmasking, 24 h post IL12-TCO-PEG_2_ injection was chosen as the time point to examine IL12 content of the tumor and plasma (Figure 5A). After isolation, tumors were weighed, cooled, and homogenized with protease inhibitors present. Because the plasma and tumors could still contain unreacted masked IL12, the samples were split and subjected ex vivo to an excess of Tz trigger **8** before being analyzed by ELISA, to assess the total amount of IL12 that potentially could have been unmasked. Figure 5B demonstrates that in plasma, almost no unmasked IL12 could be detected, whether or not diabody-Tz was pretargeted (left panel). This is consistent with the clearance pattern of the diabody and suggests no Tz was present anymore in the blood at that time point. Notably, there still was masked IL12 construct in circulation, as revealed by ex vivo Tz **8** addition. In the tumor (right panel), a similar observation was made when no diabody-Tz was pretargeted. Also here, ex vivo Tz addition revealed a high unmasking potential, indicative of IL12-construct in blood perfusing the tumor. Importantly, local IL12 unmasking in the tumor was only observed when the tumor was pretargeted with diabody-Tz. The amount of released IL12 was close to the full release potential, as demonstrated by the addition of ex vivo Tz. Two control experiments using quenching of the active Tz, or ex vivo addition of masked IL12, demonstrated that IL12 unmasking was not the result of Tz-TCO reaction during ex vivo processing of the tumor samples but instead occurred solely in vivo (Appendix A).

## 3. Discussion

We here present an effective method to selectively activate a silenced IL12 protein in the TME via bioorthogonal chemistry. Our masked IL12 strategy is based on the IEDDA pyridazine elimination reaction (aka click-to-release), which occurs between a *trans*-cyclooctene (TCO) and a tetrazine (Tz). This reaction has proven to be extremely fast, selective, and compatible with in vivo applications [29]. In this two-step approach, one of the two components functions as a linker between the PEG masks and IL12. The other component cleaves the linker to separate the two moieties, liberating the native IL12. In contrast to protease-cleavable linker systems, linker cleavage is not hampered by variable enzyme levels in the TME and in healthy tissues.

While the presented approach requires two injections instead of one for most other therapeutics, we believe the logistics will be comparable to other cancer therapies that nowadays are given in short cycles. An important advantage will be that it provides full control over the moment that therapy is triggered. The desired moment of masked IL12 injection is when the pretargeted trigger construct is largely eliminated from the circulation (e.g., <1% ID), limiting systemic activation of IL12. Sufficiently homogeneous PK of the diabody-Tz would facilitate setting a fixed interval between the two injections, simplifying the procedure in the clinic.

The conjugation of PEGs (PEGylation) has been widely used to extend the half-life of cytokines, protect against degradation, mask activity, and limit immunogenicity [40,41]. Similar to NKTR-214 (bempegaldesleukin), a clinically evaluated IL-2 cytokine inactivated by hydrolysable PEG masks [22], the click-to-release reaction in our IL12 constructs was configured for lysine chemistry. This design ensures that release of the masks returns a native lysine, rendering the reaction traceless. Masking using TCO-PEG masks could effectively silence the bioactivity of IL12 at an FG of ca. 11, while the Tz-PEG mask was far less effective at comparable functionalization. Furthermore, the quantitative release offered by the TCO linker vs. ca. 90% for the Tz-linker is a crucial parameter when 11 lysines need to be unmasked for complete gain of function. This reaction efficiency was maintained when the Tz was bound to the diabody, as the conjugate was still able to completely react with and remove all IL12 masks in vitro, returning a native human IL12 molecule with full bioactivity. While PEGylated biologicals can be subject to anti-PEG antibodies, we believe it is likely that this risk for the presented approach is relatively low, due to the use of the very short diethyleneglycol motif and the short immune system exposure caused by the fast clearance of the IL12 constructs.

Pretargeting the Tz trigger by the TAG-72-directed diabody led to efficient delivery of the trigger to the tumor. Given the potency (and potential toxicity) of IL12, a clean tumor profile of the targeting antibody and thus the Tz, is essential. In addition, in organs like the RES, a Tz-conjugated protein is likely not available extracellularly for reaction with an IL12 prodrug that is administered several hours later. Numerous radiotherapy and imaging studies have demonstrated that TAG-72 and the TAG-72 targeting CC49 mAb are very tumor-selective. Furthermore, the faster clearing CC49 diabody derivative was shown to achieve very high tumor-to-blood ratios [33]. Similarly, other slow or non-internalizing markers previously used in pretargeted radioimmunotherapy, such as CEACAM-5 or Tenascin C, would be suitable candidates for the IL12 pretargeting approach. The click histochemistry method next confirmed a substantial presence of active Tz in the tumor. When the masked IL12 was administered 24 h post-diabody-Tz, we were gratified to observe the efficient and tumor-selective IL12 activation, which we found remarkable considering that for this to occur, 11 TCO masks in one protein had to react with 11 tumor-bound Tz triggers. Further pharmacokinetics of the fully released IL12 remains to be investigated. Possibly, locally present heparin may facilitate retention of the released IL12, binding it via the cytokine’s heparin-binding domains, but the degree of washout from the tumor over time is still unknown. The present setup precludes studying the pharmacodynamics, such as CD8+ T cell influx, IFN-γ production, toxicity, and therapy, because TAG72 is a human glycoepitope not expressed in mice, requiring the use of immunocompromised mice. Also, the used human IL12 does not cross-react in mice. As a functional immune system is a prerequisite for evaluating the pharmacodynamics, compare to other IL12 therapies, and to evaluate combination therapies with e.g., checkpoint inhibitors, future studies that extend from this proof-of-principle study could employ CD34+ cell-engrafted humanized mouse models.

In summary, the work presented herein represents a novel approach to harness the therapeutic potential of IL12 through tumor-selective unmasking and activation using a pretargeted bioorthogonal trigger and click-to-release chemistry. This approach is independent of variable TME biology, and off-tissue activation is likely minimal due to the high stability of the bioorthogonally linked masks. Our method could potentially be applied to other cytokines as well (e.g. IL15, IL1), provided that lysines are critical in the binding to the cytokine’s receptor. Moreover, it will be good to explore whether a combination with other immunomodulation strategies, like checkpoint inhibitor therapy, synergizes with this approach. We believe that the proof of principle of the efficient on-tumor IL12 prodrug activation is compelling and merits further evaluation of the full anti-tumor potential of this treatment.

## 4. Materials and Methods

### 4.1. Animals and Cells

Animal studies were performed in accordance with the revised Dutch Act on Animal Experimentation and were approved by the institutional Animal Welfare Committee of the Radboud UMC (project number 2019-0037, approved on 21 December 2020). The human colon cancer LS174T cell line was obtained from ATCC (Manassas, VA, USA) and cultured in RPMI-1640 medium (Gibco/Thermo Fisher Scientific (Waltham, MA, USA)) supplemented with 2 mM glutamine (Gibco) and 10% fetal calf serum (FCS; Sigma Aldrich (St. Louis, MA, USA)). Before harvesting, cells were grown to >50% confluency and detached using trypsin EDTA (Gibco). The HEK-Blue-IL12 reporter cell was obtained from Invivogen (San Diego, CA, USA) and cultured in DMEM medium (Gibco), containing 4.5 g/L glucose and Glutamax, supplemented with 10% heat-inactivated fetal calf serum. Before harvesting, the cells were grown to >80% confluency and detached using gentle slamming of the culture flasks. Female BALB/c nude mice (7–9 week old, 18–22 g body weight (Janvier, Saint-Berthevin, France)) were subcutaneously inoculated ca. 1.5 × 10^6^ LS174T cells (in 100 µL serum-free culture medium), high on the hind limb. Tumor size was determined by caliper measurements in 3 dimensions (tumor volume = ½ × l × w × h) twice per week. Biodistribution and kinetics studies started when the tumors reached 0.25–0.30 cm^3^. At the end of the studies, the animals were euthanized by CO_2_ asphyxiation (gradual fill), blood was obtained by cardiac puncture and organs and tissues of interest were harvested, blotted dry, weighed, and the sample radioactivity was measured in a shielded well-type γ-counter together with a ten-plo aliquots of the injected dose to calculate %ID/g or %ID (where indicated). Stomachs and intestines were not emptied before γ-counting.

### 4.2. SDS-Polyacrylamide Gel Electrophoresis (SDS-PAGE)

SDS-PAGE on samples was performed under non-reducing conditions on a Mini-PROTEAN Tetra Cell system using 4–20% Mini-PROTEAN TGX Precast Protein Gels (BioRad Laboratories (Hercules, CA, USA)), with Precision Plus Protein All Blue protein standards (BioRad Laboratories) according to the manufacturer’s instructions. Per sample ca. 1.5 µg protein was loaded. The gels were stained with Coomassie Brilliant Blue (Sigma Aldrich) for protein detection. Diabody-Tz and TCO-PEG_2_-masked IL12 constructs were radiolabelled with iodine-125 using Bolton–Hunter reagent (Pierce/Thermo Fisher Scientific (Waltham, MA, USA)), as previously described [42]. For unmasking experiments, samples were incubated with an excess of trigger (5 eq relative to the linker, 30 min, 37 °C) prior to gel electrophoresis.

### 4.3. IL12 Bioassay

Native IL12 or IL12-PEG conjugates were plated in a round-bottom plate (Costar (Arlington, VA, USA)) and reacted with the indicated trigger compound in serum-free culture medium (DMEM, 1 h, 37 °C). Next, a 1:2-step dilution was made, after which 80.000 reporter cells (HEK-Blue IL12 cells; InvivoGen) were added in complete medium containing 10% FCS. IL12 bioactivity was determined after 20–22 h, based on colorimetric quantification of Secreted Alkaline Phosphatase (SEAP) using the reagent QUANTI-Blue (InvivoGen) at OD630 nm. In the experiments where conjugated diabody was used, incubations were completed in complete medium with 10% FCS to block nonspecific interactions with plastic. Blocking results in a general shift of the curves to the left.

### 4.4. Blood Kinetics and Biodistribution Analysis

To assess their blood kinetics, diabody-Tz and TCO-PEG_2_-masked IL12 constructs were radiolabelled with iodine-125 using Bolton-Hunter reagent, as previously described [42], and injected i.v. in tumor-free mice (IL12: 10 µg, ca. 0.4 MBq in 100 µL PBS per mouse; diabody: 40 µg, ca. 0.4 MBq in 100 µL PBS per mouse; *n* = 4). Detection of radioactivity was done in blood samples collected via the vena saphena at various timepoints (IL12: 2, 30, 60 min, 3, 6, 24, and 72 h post-injection; diabody: 1, 2, 6, 24, 48, 72 h post-injection). Weight of the blood samples and counts as measured by a gamma-counter (Wizard 1480, PerkinElmer (Waltham, MA, USA)) were used to calculate the percentage of the injected dose per gram (%ID/g).

For biodistribution analysis, diabody-Tz was radiolabelled with I-125 using Bolton–Hunter reagent, and injected i.v. in tumor-bearing mice (150 µg, ca. 0.7 MBq in 100 µL PBS per mouse; *n* = 4). After 48 h post-injection, organs were isolated and weighed. Gamma counts were used to calculate the percentage of the injected dose per gram (%ID/g), whereas some organs were not corrected for weight, reporting values in %ID.

### 4.5. ELISA Detection of IL12 in Tumors and Plasma After Tz Trigger Pretargeting

Diabody-Tz **12** was injected i.v. in mice bearing LS174T tumors (40 µg in 100 µL PBS per mouse; *n* = 4). Twenty-four hours later, when the diabody was nearly completely cleared from blood, TCO-masked IL12 **10** (FG 11) was injected i.v. in selected groups (10 µg in 100 µL PBS) and the mice were euthanized twenty-four hours later. Tumors were isolated and collected in MagNA lyser green bead tubes (Roche/Thermo Fisher Scientific (Waltham, MA, USA)) on ice and added with protease inhibitor cocktail (1 tablet in 5mL PBS: 2 mL/g tumor; Roche) followed by homogenization and debris removal. Blood was collected in heparinized vials on ice, plasma was separated, and 2-fold diluted with protease inhibitor cocktail.

In one control experiment (See Appendix A), tumors were isolated at 26 h post-diabody injection, split in half, and collected in different MagNA lyser green bead tubes on ice. To one-half a 100 eq molar excess of TCO **3** in protease inhibitor solution was added to deactivate all diabody-Tz present. This was done to exclude unmasking due to Tz activity during the sample work-up. The other half received only a protease inhibitor cocktail. Subsequently, the samples were homogenized.

In another control experiment (See Appendix A), some tumors did not receive masked cytokine in vivo. After these tumors were isolated at 26 h post-diabody injection, 1 ng/mL IL12-TCO-PEG_2_ in PBS was added ex vivo to verify retained trigger potency.

Detection of unmasked IL12 constructs in the tumor homogenates and plasma was carried out by ELISA (‘sandwich setup’, IL12p70 Human ELISA kit, Invitrogen/Thermo Fisher Scientific (Waltham, MA, USA)). For this, the samples were 9-fold diluted in the kit’s ELISPOT buffer, with or without addition of a large molar excess of small-molecule tetrazine trigger (**8**) to unmask remaining IL12-TCO-PEG_2_ constructs ex vivo (37 °C for 2 h). A total of 100 µL of the mixtures was transferred to an anti-hIL12 p70-coated ELISA plate and incubated for 18 h at 4 °C. Consecutive steps were carried out according to the manufacturer’s instructions.

### 4.6. Click-Histochemistry and Autoradiography to Detect In Vivo Tetrazine

Tumor-bearing mice were injected with ^125^I-labeled diabody-Tz **12** (150 μg, 0.7 MBq). After 48 h, tumors were isolated and directly frozen at −80 °C for autoradiography and Tz staining. Frozen tumor sections (5 μm) were fixated for 10 min in ice-cold acetone (−20 °C). Consecutive sections were used for autoradiography and Tz staining. For autoradiography, slides were exposed to a Fujifilm BAS cassette 2025 (Fuji (Tokyo, Japan)) for 14 days. Phospholuminescence plates were scanned using a Fuji BAS-1800 II bioimaging analyzer (Fuji) at a pixel size of 25 × 25 μm. Images were analyzed with Aida Image Analyzer software (Raytest, version 5.1 SP6). TCO-Tz click chemistry was used to stain tetrazines in the tumor sections. Between consecutive steps of the staining process, sections were washed 3 times in PBS. Endogenous biotin/avidin was blocked using Avidin/Biotin Blocking Kit (SP-2001, Vector Laboratories (Newark, CA, USA)), after which slides were incubated with TCO-PEG_4_-biotin (BP-23847, Broadpharm (San Diego, CA, USA)) (0.1 μg/mL in 1%BSA/PBS, overnight, RT). Endogenous peroxidase activity was blocked with 0.3% H_2_O_2_ (Sigma Aldrich) (diluted in PBS, 10 min, RT) before incubation with VECTASTAIN^®^ Elite^®^ ABC-HRP Kit (PK-6100, Vector Laboratories) (in 1% BSA/PBS, 30 min, RT). Finally, 3-3′-diaminobenzidine (BrightDAB, Immunologic (Arnhem, The Netherlands)) was used to visualize peroxidase activity in the sections. Slides were counterstained with hematoxylin, dehydrated, and mounted with a cover slip using Permount™ (Thermo Fisher Scientific).

Additional materials and methods are included in the Appendix A.

## Figures and Tables

**Figure 1 pharmaceuticals-18-01380-f001:**
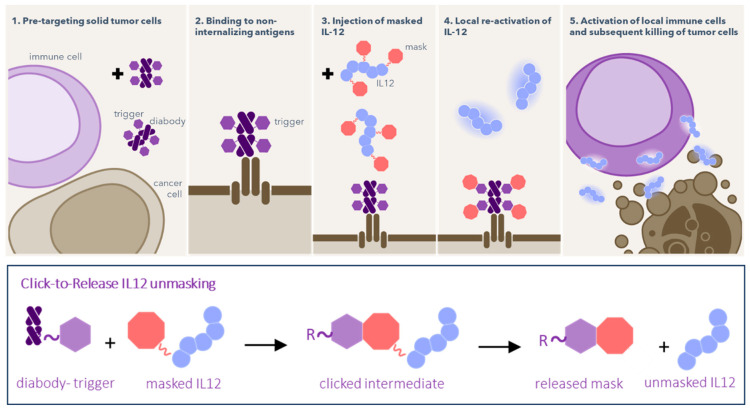
Schematic of pretargeted unmasking of IL12, consisting of two components that are injected separately. First, the “trigger” compound (purple hexagons) is pretargeted by a conjugated diabody to a non-internalizing tumor target. When the trigger has cleared from blood, masked IL12 is administered, chemically silenced by PEG masks bound via a click-to-release linker (red octagons). Once the cytokine passes the tumor, the trigger and linker react, releasing the masks to afford native IL12 in the tumor, which can then activate the immune system. The trigger is either a Tz with the linker being a TCO, or vice versa.

**Figure 2 pharmaceuticals-18-01380-f002:**
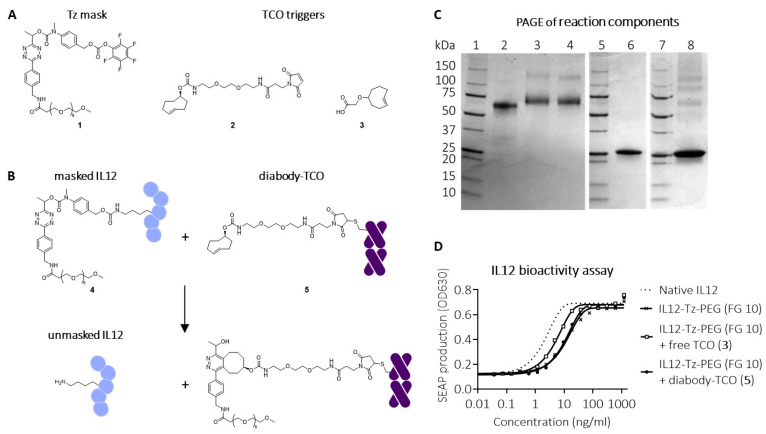
Click-to-release IL12 employing a Tz linker and a TCO trigger. (**A**) Structures of conjugatable Tz-PEG mask (**1**), conjugatable TCO trigger (**2**), and free TCO trigger (**3**). (**B**) Reaction between corresponding masked IL12-Tz-PEG (**4**) and diabody-TCO conjugate (**5**), affording unmasked IL12. (**C**) SDS-PAGE analysis of I-125-labelled native IL12 (lane 2), IL12-Tz-PEG **4** (lane 3), IL12-Tz-PEG **4** with addition of free TCO **3** (lane 4), I-125-labelled diabody-TCO **5** (Lane 6), I-125-labelled IL12-Tz-PEG **4** with addition of I-125-labelled diabody-TCO **5** (lane 8); Lanes 1, 5, 7: markers. (**D**) Bioactivity assay of IL12-Tz-PEG **4,** with functionalization grade (FG) 10 and native IL12, following reaction with free TCO **3** or diabody-TCO **5**, showing partial masking and unmasking.

**Figure 3 pharmaceuticals-18-01380-f003:**
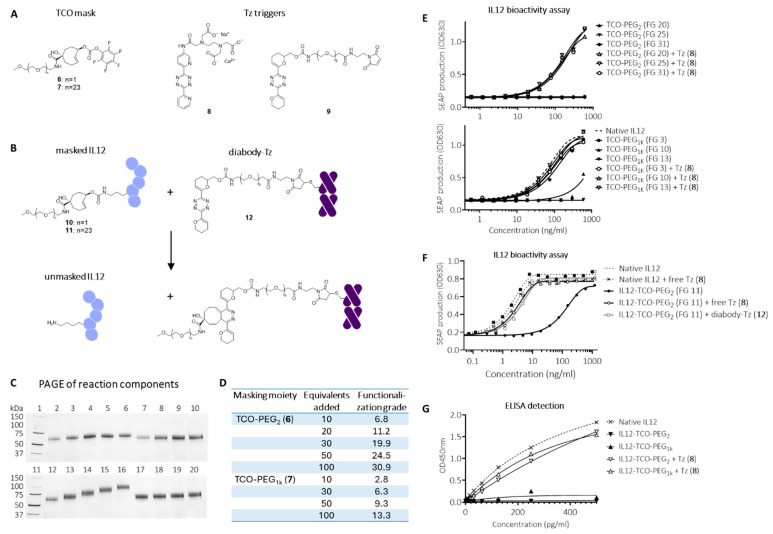
Click-to-release IL12 employing a TCO linker and a Tz trigger. (**A**) Structures of conjugatable TCO-PEG masks (**6** and **7**), free Tz trigger (**8**), and conjugatable Tz trigger (**9**). (**B**) Reaction between corresponding masked IL12-TCO-PEG constructs **10** and **11** and diabody-Tz conjugate **12,** affording unmasked IL12. (**C**) SDS-PAGE analysis of I-125 labelled native IL12 (lanes 2 and 12), IL12-TCO-PEG_2_ constructs (**10**) conjugated by adding 10, 30, 50, or 100 equivalents of mask **6** relative to IL12 (lanes 3 to 6), and the same compounds after addition of Tz **8** (5 eq relative to TCO; lanes 7 to 10), IL12-TCO-PEG_1K_ constructs (**11**) conjugated by adding 10, 30, 50, or 100 equivalents of mask **7** (Lanes 13 to 16), and the same compounds after addition of Tz **8** (5 eq relative to TCO; lanes 17 to 20). (**D**) IL12-mask functionalization grades (FG) determined by ^111^In-labeled Tz-DOTA analysis. (**E**) HEK-Blue IL12 reporter assay performed on IL12-TCO-PEG_2_ (**10**; top) and -PEG_1K_ (**11**; bottom) constructs and native IL12, with (open symbols) and without (filled symbols) activation by Tz **8**. (**F**) HEK-Blue IL12 reporter assay with IL12-TCO-PEG_2_ **10** with FG of 11.2, with (open symbols) and without (filled symbols) activation by Tz **8**, or diabody-Tz conjugate **12**. (**G**) hIL12-p70 ELISA assay detection of IL12-TCO-PEG_2_ or IL12-TCO-PEG_1K_ constructs with FG of resp. 24.5 and 13.3 without (filled symbols) and with (open symbols) activation by Tz **8**.

**Figure 4 pharmaceuticals-18-01380-f004:**
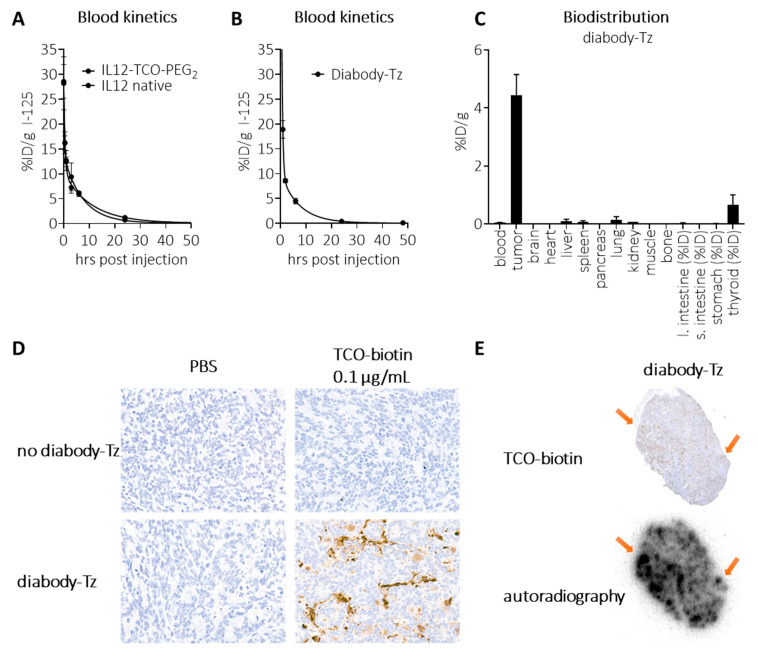
In vivo characterization of diabody-Tz **12** and IL12-TCO-PEG_2_
**10**. (**A**) Blood kinetics of I-125 labelled IL12 and IL12-TCO-PEG_2_ (FG 11), administered i.v. in tumor-free mice (10 µg/mouse), followed by detection of radioactivity in blood samples collected at various timepoints (2, 30, 60 min, 3, 6, 24, and 72 h post injection). (**B**) Blood kinetics of I-125-labelled diabody-Tz **12**, administered i.v. in tumor-free mice (40 µg/mouse), followed by detection of radioactivity in blood samples collected at various timepoints (2, 30, 60 min, 3, 6, 24, and 72 h post injection). (**C**) Biodistribution of I-125-labelled diabody-Tz **12**, administered i.v. in LS174T-tumor bearing mice (150 µg/mouse), followed by detection of radioactivity in various tissues at 48 h post-injection. (**D**) For selected groups, the tumor from a representative mouse was isolated and snap-frozen in OCT matrix, and the presence of reactive Tz in the tumor was detected by TCO-PEG_4_-biotin on 5 μm tissue slides. (**E**) I-125 Autoradiography of the same tumor slices as panel (**D**) with arrows indicating areas where clusters of Tz are visible. Data represent the mean percentage injected dose (% ID) or injected dose per gram (% ID/g) with s.d. (*n* = 4).

**Figure 5 pharmaceuticals-18-01380-f005:**
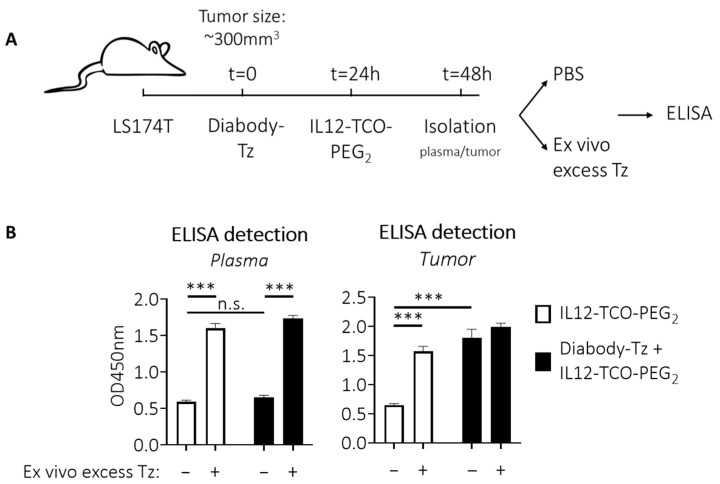
In vivo IL12 detection. (**A**) Schematic workflow to detect (unmasked) IL12 in tumors and plasma after trigger pretargeting. Diabody-Tz was injected i.v. in mice bearing LS174T tumors (40 µg in 100 µL PBS per mouse; *n* = 4). Twenty-four hours later, when the diabody was virtually cleared from blood, TCO-masked IL12 was injected i.v. (FG 11; 10 µg in 100 µL PBS). Twenty-four hours later, tumors and blood were isolated and processed with protease inhibitor cocktail present. Before the ELISA, some samples received an excess of Tz **8** to release all remaining masks. (**B**) ELISA analysis showing no IL12 unmasking in plasma in vivo due to low blood level of Tz at the time of IL12-TCO-PEG injection (left panel). Right panel shows that pre-localized Tz in the tumor is able to unmask the IL12 construct locally. Data represent mean OD450nm values with SEM (*n* = 4). Significance: n.s. indicates *p* > 0.05; *** indicates *p* ≤ 0.001.

## Data Availability

The data that support the findings reported herein are available on reasonable request from the corresponding author.

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
