# Peer review of "Click-to-Release for Controlled Immune Cell Activation: Tumor-Targeted Unmasking of an IL12 Prodrug"

_pharmaceuticals, 2025, doi:10.3390/ph18091380_

Round 1

Reviewer 1 Report

Comments and Suggestions for Authors

In the present manuscript, the authors have come up with a novel controlled immune cell activation for IL12 prodrug for anticancer therapy. In general, the manuscript is well written and the research is extensive. Further, the authors are suggested to consider following comments to improve the manuscript quality and details.

  • The study employs BALB/c nude mice, which lack a fully functional immune system. Given that IL12 is a potent immune-modulating cytokine, its therapeutic efficacy and toxicity profile cannot be fully evaluated in such models. Future studies should consider using immunocompetent or humanized mouse models to better assess immune activation, cytokine dynamics, and tumor response. Please comment on the same on extrapolation of mouse models to humans.
  • While tumor-localized unmasking of IL12 is demonstrated, the manuscript does not explore the duration of IL12 retention or its functional persistence within the tumor microenvironment. Incorporating time-course studies of IL12 levels and downstream immune markers (e.g., IFN-γ, CD8+ T cell infiltration) would strengthen the mechanistic understanding and therapeutic relevance. Please comment on this aspect.
  • The study shows blood clearance of the diabody and IL12 constructs but does not evaluate potential off-target activation in key immune organs such as the liver, spleen, or lymph nodes. Including biodistribution and ELISA data from these organs would help confirm tumor specificity and rule out unintended systemic activation.
  • PEGylation is known to induce anti-PEG antibodies, particularly with repeated dosing, which may affect pharmacokinetics and safety. The manuscript may discuss immunogenicity risks and consider alternative masking strategies or mitigation approaches for clinical translation.
  • Although IL12 bioactivity is restored post-unmasking, the study may be improved with data on immune cell activation, cytokine release, or tumor regression. Including flow cytometry or immunohistochemistry data to assess immune cell recruitment and activation would provide functional validation of the therapeutic concept.

Author Response

Reviewer 1

In the present manuscript, the authors have come up with a novel controlled immune cell activation for IL12 prodrug for anticancer therapy. In general, the manuscript is well-written and the research is extensive. Further, the authors are suggested to consider following comments to improve the manuscript quality and details.

Comment 1.1

The study employs BALB/c nude mice, which lack a fully functional immune system. Given that IL12 is a potent immune-modulating cytokine, its therapeutic efficacy and toxicity profile cannot be fully evaluated in such models. Future studies should consider using immunocompetent or humanized mouse models to better assess immune activation, cytokine dynamics, and tumor response. Please comment on the same on extrapolation of mouse models to humans.

Response 1.1

We agree with the reviewer that studying the bioactivity of IL12 is not possible in immunocompromised mice and that future studies extending from the proof of principle of the present paper should include such models. To achieve this, a few aspects need to be considered. Most importantly, human IL12, as is used in our constructs, does not cross-activate mouse IL12 receptors. The immune system from immunocompetent mice therefore would not react to our therapy. Moreover, the TAG-72 target we used to localize IL12 release is a human target, which therefore is clinically relevant, but unfortunately only present in human tumor cells. As it is a glycoepitope, it is also very difficult to transfect into a mouse tumor cell line. Only by completely rebuilding the system by masking mouse IL12, using a mouse tumor and targeting a mouse tumor antigen the bioactivity could be evaluated in non-humanized immunocompetent mice. Here, however, its translational value remains questionable, as masking and unmasking of murine IL12 by conjugation to lysines could be fairly different due to differences with the human genomic sequence.

Alternatively, it could indeed be a good follow up strategy to perform future follow-up studies in humanized mouse models with long-lived humanization (CD34+ stem cell transfer). Released IFN-γ could be a measure for activation of the transplanted immune system. Potentially, also differences in tumor growth could be detected in such systems.

We have now briefly addressed this aspect in the discussion. [Discussion section. Paragraph 5]

Comment 1.2

While tumor-localized unmasking of IL12 is demonstrated, the manuscript does not explore the duration of IL12 retention or its functional persistence within the tumor microenvironment. Incorporating time-course studies of IL12 levels and downstream immune markers (e.g., IFN-γ, CD8+ T cell infiltration) would strengthen the mechanistic understanding and therapeutic relevance. Please comment on this aspect.

Response 1.2

We agree with the reviewer that follow up studies should explore the duration of IL12 retention and its functional persistence within the tumor. Radiolabeling the masked IL12 would allow us to follow its tumor uptake, but this signal would not differ between masked and unmasked state, as the label is bound to the protein. Overlaying this data with the ELISA method to detect (un)masked IL12 over time would show us how long active IL12 stays present in the tumor. To also evaluate immune activation and functional persistence over time, measuring  IFN-γ release and/or CD8+ T cell infiltration would indeed both be good additional parameters to assess. Here we however encounter the same considerations as discussed under comment 1. Using CD34+ cell humanized mice or a full mouse system in future follow-up studies could potentially tackle these issues. We believe the above is best suited for a follow up manuscript that extends from the proof of principle presented in the present manuscript.

As this topic is very related to the topic in comment 1, we have now discussed it at the same location. [Discussion section. Paragraph 5]

Comment 1.3

The study shows blood clearance of the diabody and IL12 constructs but does not evaluate potential off-target activation in key immune organs such as the liver, spleen, or lymph nodes. Including biodistribution and ELISA data from these organs would help confirm tumor specificity and rule out unintended systemic activation.

Response 1.3

The reviewer touches upon an important aspect, which in our opinion is partly covered in our manuscript. Toxicity by IL12 would either be indirectly caused by the mediators secreted by the immune system or (in the minority of cases) directly by the few non-immune tissues that express low levels of the IL12 receptor (e.g. endothelial cells fibroblasts, glia cells). In all these cases, however, one would need a responsive IL12 receptor, which is absent in our setting.

The activation of masked IL12 is triggered by the presence of a trigger-loaded diabody. The specificity of this chemical reaction, and the superior in vivo stability of the TCO linker is shown before in many of our and the referenced papers. Furthermore, it is illustrated by figure 5, where the IL12 activation remains low when no trigger is provided. We therefore believe that to answer this question one should focus on the remarkably low diabody-trigger biodistribution to other tissues than the tumor (Figure 4c). In addition, the diabody-Tz is likely not extracellularly available for reaction in organs like the RES. In absence of a responsive IL12 receptor readout, the clean biodistribution of the trigger is a strong indication that off-target toxicity will likely be low.       

Comment 1.4

PEGylation is known to induce anti-PEG antibodies, particularly with repeated dosing, which may affect pharmacokinetics and safety. The manuscript may discuss immunogenicity risks and consider alternative masking strategies or mitigation approaches for clinical translation.

Response 1.4

Anti-PEG antibodies indeed form a relevant development risk for biologicals. We believe the chances these antibodies will develop in our approach are low, since we use extremely short PEG chains (e.g. PEG2, which can hardly be called PEG). Moreover, the IL12-TCO-PEG will clear fast, which limits the exposure to the immune system, thereby reducing the time the immune can develop such responses. Finally, after on tumor reaction and clearance from systemics, the PEG chains will have been taken over by the diabody still bound specifically to the tumor (see reaction schematic). This will significantly limit the time a potential PEG-directed ADA can bind and neutralize the active component, IL12. Hence, we do not expect that ADAs form a prominent development risk.

We have now discussed these topics more extensively in the discussion. [Discussion section, Paragraph 3]  

Comment 1.5

Although IL12 bioactivity is restored post-unmasking, the study may be improved with data on immune cell activation, cytokine release, or tumor regression. Including flow cytometry or immunohistochemistry data to assess immune cell recruitment and activation would provide functional validation of the therapeutic concept.

Response 1.5

We agree with the reviewer on the relevance of functional and therapeutic validation of our concept. Please refer to the responses to the reviewer’s previous comments regarding the absence of these evaluations in the present manuscript.   

Reviewer 2 Report

Comments and Suggestions for Authors
  1. The concept of using a click-to-release system for selective IL12 activation in tumors is innovative and addresses a critical clinical challenge of cytokine therapy toxicity. The authors should better emphasize how their approach compares to existing IL12 delivery strategies, such as nanoparticle encapsulation, fusion proteins, or tumor-responsive linkers.
  2. The abstract outlines the masking/unmasking principle, but it would benefit from a clearer mechanistic explanation of how PEGylation affects IL12’s bioactivity and how the IEDDA pyridazine elimination reaction achieves complete restoration of native function. Including more detail on reaction kinetics and stability in physiological conditions would strengthen the rationale.
  3. The authors use TAG-72 targeting for prelocalization. They should discuss the clinical relevance of TAG-72 expression profiles across tumor types and whether this approach could be adapted for other tumor antigens.
  4. The two-step administration (diabody-tetrazine followed by masked IL12) raises practical considerations for clinical translation. The authors should discuss patient workflow feasibility, potential immunogenicity of the diabody, and how timing between injections might vary in humans.
  5. It would be useful to clarify whether the authors compared the click-to-release IL12 prodrug against conventional IL12 dosing or other site-specific delivery systems in vivo, to contextualize improvements in safety and efficacy.
  6. The proof-of-principle is promising, but the authors could strengthen the impact by outlining next steps, such as scaling to larger animal models, combining with checkpoint inhibitors, or applying the platform to other potent cytokines or immunostimulatory agents.

Author Response

Reviewer 2

Comment 2.1

The concept of using a click-to-release system for selective IL12 activation in tumors is innovative and addresses a critical clinical challenge of cytokine therapy toxicity. The authors should better emphasize how their approach compares to existing IL12 delivery strategies, such as nanoparticle encapsulation, fusion proteins, or tumor-responsive linkers.

Response 2.1

Although a bit condensed, each of these strategies was discussed in the second paragraph of the introduction. To make our potential differentiation more clear we have added a line at the end of the introduction.

Comment 2.2

The abstract outlines the masking/unmasking principle, but it would benefit from a clearer mechanistic explanation of how PEGylation affects IL12’s bioactivity and how the IEDDA pyridazine elimination reaction achieves complete restoration of native function. Including more detail on reaction kinetics and stability in physiological conditions would strengthen the rationale.

Response 2.2

Although there is limited space in the abstract, we have now sharpened the existing wording to cover the suggested topics. [Abstract, objectives]

Comment 2.3

The authors use TAG-72 targeting for pre-localization. They should discuss the clinical relevance of TAG-72 expression profiles across tumor types and whether this approach could be adapted for other tumor antigens.

Response 2.3

We have now added more info on the clinical relevance of TAG-72 and the adaptability to other antigens. [Results section, paragraph 2; Discussion section, paragraph 4]

Comment 2.4

The two-step administration (diabody-tetrazine followed by masked IL12) raises practical considerations for clinical translation. The authors should discuss patient workflow feasibility, potential immunogenicity of the diabody, and how timing between injections might vary in humans.

Response 2.4

We have now discussed this more elaborately in the discussion. [Discussion section, paragraph 2]

Comment 2.5

It would be useful to clarify whether the authors compared the click-to-release IL12 prodrug against conventional IL12 dosing or other site-specific delivery systems in vivo, to contextualize improvements in safety and efficacy.

Response 2.5

We agree with the reviewer that these are important experiments. However, to make a robust comparison with conventional IL12 dosing or other delivery systems with respect to toxicity and efficacy one would need a functional immune system. Activation of the immune system cannot be studied in immunocompromised mice. Using non-humanized immunocompetent mice is not feasible because human IL12, as is used in our constructs, does not cross-activate mouse IL12 receptors. The immune system from immunocompetent mice therefore would not react to our therapy. Moreover, the TAG-72 target we used to localize IL12 release is a human target, which therefore is clinically relevant, but unfortunately only present in human tumor cells. As it is a glycoepitope, it is also very difficult to transfect into a mouse tumor cell line. The use of a human tumor dictates we must use immunocompromised mice. Only by completely rebuilding the system by masking mouse IL12, using a mouse tumor and a mouse targeting device the bioactivity could be evaluated in non-humanized immunocompetent mice. Here, however, it’s translational value remains questionable, as masking and unmasking of murine IL12 by conjugation to lysines could be fairly different due to differences in the genomic sequence. Alternatively, a good follow up strategy would be to perform future studies in mouse models with long-lived humanization (CD34+ stem cell transfer) and compare our approach with conventional IL12 dosing. Released IFN-γ could be a measure for activation of the transplanted immune system. Potentially, also differences in tumor growth could be detected in such systems.

We have now discussed this more elaborately in the discussion, aligning with the adaptions made in response to the comments from reviewer 1. [Discussion section. Paragraph 5]  

Comment 2.6

The proof-of-principle is promising, but the authors could strengthen the impact by outlining next steps, such as scaling to larger animal models, combining with checkpoint inhibitors, or applying the platform to other potent cytokines or immunostimulatory agents.

Response 2.6

We have now added some of these next steps to the discussion. [Discussion section, paragraph 6]

Reviewer 3 Report

Comments and Suggestions for Authors

This manuscript details the development of a bio-orthogonal masking approach that allows for the use of IL12 to target cancer cells. IL12 is not useful as a tumor treatment due to associated toxicity, but this may be overcome by targeting the protein to the desired tumor cell. The authors explore several approaches where the IL12 is masked by PEG chains linked to the protein through a carbamate linker that can be cleaved by a click reaction with a tetrazine. After some optimization, it appears this approach may be workable, and some significant results are presented that show that the system may be useful and deserves further study. Overall the manuscript is well written, but I did find a couple of minor issues that I think could be addressed before publication, including the following:

-Page 2, paragraph 2, "Masked cytokines have been developed designed to be activated..." shouldn’t it be developed or designed, one or the other?

-Figure 2D is very small and therefore difficult to read, perhaps it could be reworked to be larger and more legible?

Author Response

Reviewer 3

Comment 3.1

Page 2, paragraph 2, "Masked cytokines have been developed designed to be activated..." shouldn’t it be developed or designed, one or the other?

Response 3.1

We thank the reviewer for spotting this; we have corrected the typo.

Comment 3.2

Figure 2D is very small and therefore difficult to read, perhaps it could be reworked to be larger and more legible?

Response 3.2

We have now enlarged the figure and increased the font size.